# Optimizing the Dosing Regimens of Daptomycin Based on the Susceptible Dose-Dependent Breakpoint against Vancomycin-Resistant Enterococci Infection

**DOI:** 10.3390/antibiotics8040245

**Published:** 2019-11-29

**Authors:** Wichai Santimaleeworagun, Dhitiwat Changpradub, Sudaluck Thunyaharn, Jatapat Hemapanpairoa

**Affiliations:** 1Department of Pharmacy, Faculty of Pharmacy, Silpakorn University, Nakorn Pathom 73000, Thailand; 2Pharmaceutical Initiative for Resistant Bacteria and Infectious Diseases Working Group (PIRBIG), Nakorn Pathom 73000, Thailand; 3Division of Infectious Disease, Department of Medicine, Phramongkutklao Hospital, Bangkok 10400, Thailand; 4Faculty of Medical Technology, Nakhonratchasima College, Nakhon Ratchasima 30000, Thailand; 5Department of Pharmacy Practice and Pharmaceutical Care, Faculty of Pharmaceutical Sciences, Burapha University, Chonburi 20131, Thailand

**Keywords:** daptomycin, *Enterococcus faecium*, MIC, Monte Carlo simulation, VRE

## Abstract

Daptomycin, a lipopeptide antibiotic, is one of the therapeutic options used for the treatment of vancomycin-resistant enterococci (VRE). Recently, the Clinical and Laboratory Standards Institute (CLSI) M100 30th edition has removed the susceptibility (S) breakpoint for *Enterococcus faecium* and replaced it with a susceptible dose-dependent (SDD) breakpoint of ≤4 μg/mL, with a suggested dosage of 8–12 mg/kg/day. Herein, we aimed to determine the minimum inhibitory concentration (MIC) values of daptomycin against clinical VRE isolates and to study the appropriate daptomycin dosing regimens among critically ill patients based on the new susceptibility CLSI breakpoint. The MIC determination of daptomycin was performed using E-test strips among clinical VRE strains isolated from patients at the Phramongkutklao Hospital. We used Monte Carlo simulation to calculate the probability of target attainment (PTA) and the cumulative fraction of response (CFR) of the ratio of the free area under the curve to MIC (*f*AUC_0–24_/MIC) > 27.4 and *f*AUC_0–24_/MIC > 20 for survival and microbiological response, respectively, at the first day and steady state. Further, we determined that the simulated daptomycin dosing regimen met the minimum concentration (Cmin) requirements for safety of being below 24.3 mg/L. All of the 48 VRE isolates were *E. faecium* strains, and the percentiles at the 50th and 90th MIC of daptomycin were 1 and 1.5 μg/mL, respectively. At MIC ≤ 2 μg/mL, a daptomycin dosage of 12 mg/kg/day achieved the PTA target of survival and microbiological response at the first 24 h time point and steady state. For a MIC of 4 μg/mL, none of the dosage regimens achieved the PTA target. For CFR, a dosage of 8–12 mg/kg/day could achieve the 90% CFR target at the first day and steady state. All dosing regimens had a low probability of Cmin being greater than 24.3 mg/L. In conclusion, the MIC of VRE against daptomycin is quite low, and loading and maintenance doses with 8 mg/kg/day were determined to be optimal and safe.

## 1. Introduction

The prevalence of vancomycin-resistant enterococci (VRE), a nosocomial pathogen, is increasing, especially for *Enterococcus faecium*. Invasive VRE infections are commonly found among intensive care unit (ICU) patients and are associated with high mortality and long-term hospitalization [1,2,3]. Daptomycin and linezolid are first-line antibiotics for treating VRE infections [4,5,6,7]. Daptomycin, a lipopeptide antibiotic, plays an important role in the treatment of serious infections or infections in immunocompromised patients requiring bactericidal antibiotics [8,9].

Currently, daptomycin is approved for treatment of complicated skin and soft tissue infections as well as *Staphylococcus aureus* bacteremia with right-side endocarditis. Daptomycin is clinically used for VRE treatment; however, a standard dose (4–6 mg/kg/day) of daptomycin for VRE bloodstream infection has been shown to result in a poorer survival rate than a high dose of ≥9 mg/kg/day [10,11,12]. Moreover, the Clinical and Laboratory Standards Institute (CLSI) M100 30th edition has removed the susceptibility (S) breakpoint for *E. faecium* and replaced it with a minimum inhibitory concentration (MIC) susceptible dose-dependent (SDD) breakpoint of ≤4 μg/mL. Regarding this new susceptibility breakpoint for enterococci, CLSI suggests a daptomycin dosing regimen of 8–12 mg/kg/day, while for other enterococci species, the susceptible MIC breakpoint at 2 μg/mL remains in place, with a recommended daptomycin dose of 6 mg/kg/day [13,14].

Daptomycin has concentration-dependent bactericidal activity and its pharmacokinetic and pharmacodynamic (PK/PD) target for efficacy is the ratio of the area under the curve to MIC (AUC_0–24_/MIC) or the ratio of the free area under the curve to MIC (*f*AUC_0–24_/MIC) [15,16]. According to the SDD breakpoint with a high dose of daptomycin, VRE treatment has to achieve its PK/PD target to reduce the risk of underdosing daptomycin. In the same way, a high dose of daptomycin may increase the risk of creatine phosphokinase (CPK) elevation, and musculoskeletal toxicity is associated with doses at the minimum concentration (Cmin) [10,17]. In critically ill patients, an alteration in drug pharmacokinetics can occur [18,19,20], so optimal dosing of daptomycin is of great concern. The purpose of this study is to assess the in vitro activity of daptomycin as well as to evaluate dosing regimens of daptomycin in critically ill patients based on its PK/PD target for efficacy and safety against VRE isolates.

## 2. Results

### 2.1. Pharmacodynamic Profiling

All of the 48 studied VRE clinical isolates were *E. faecium* strains. The MIC_50_ and MIC_90_ for vancomycin were 128 and >128 μg/mL, respectively. Daptomycin MIC values against VRE isolates ranged from 0.38 to 4 μg/mL. The MIC_50_ and MIC_90_ values were 1 and 1.5 μg/mL, respectively. Daptomycin resistance among VRE isolates was not observed (Figure 1).

### 2.2. Pharmacokinetic and Pharmacodynamic Analysis and Dosing Simulations

The probability of target attainment (PTA) of daptomycin dosing regimens at specific MICs with target fAUC_0–24_/MIC > 27.4 and fAUC_0–24_/MIC > 20 during the first 24 h and at the steady state are shown in Figure 2 and Figure 3, respectively. Target attainment for all regimens during the first 24 h was lower than steady state. None of the dosing regimens exceeded 90% for a MIC of 4 μg/mL. The dosing regimen of 12 mg/kg every 24 h of daptomycin gave the target attainment of fAUC_0–24_/MIC > 27.4, which exceeded 90% for a MIC of ≤2 μg/mL during first 24 h and at steady state, respectively. For daptomycin at a MIC of 1 μg/mL, dosing of 6 mg/kg/day exceeded the 90% PTA of fAUC_0–24_/MIC targets. All daptomycin dosing regimens gave a Cmin of daptomycin below 24.3 mg/mL (Table 1).

For VRE, the cumulative fraction of response (CFR) of fAUC_0–24_/MIC > 27.4 and 20 exceed 90% with 8–12 mg/kg daptomycin dosing during the first 24 h and at steady state, whereas a 6 mg/kg dosing had a CFR of fAUC_0–24_/MIC > 20 exceeding 90% (Table 1).

## 3. Discussion

This is the first study in Thailand that has tested the in vitro susceptibility of daptomycin against VRE. Even though VRE are not the major multidrug-resistant pathogens in Thailand, their incidence is increasing and bacteremia is most common, especially in the ICU or immunocompromised patients [1,21]. 

The activity of daptomycin against enterococci was not affected by vancomycin resistance. Previous studies have shown that the daptomycin MIC of *E. faecium* isolates is higher than that of *Enterococcus faecalis* [22,23]. However, all of the VRE isolates here were *E. faecium*, and no daptomycin-resistant VRE were observed, and daptomycin showed better activity against VRE strains compared with data from a systematic review. The MIC_90_ of daptomycin was lower than that reported from other studies (1.5 vs. 2–4 μg/mL). In the case of daptomycin, MICs of 2 and 4 μg/mL were isolated from urine and blood, respectively, with no history of daptomycin treatment. None had used daptomycin before VRE isolates, but they documented prior use of vancomycin. A correlation between reduced susceptibility to daptomycin and vancomycin was found in *S. aureus*, but such a correlation was not found in this study [24]. Reduced susceptibility to daptomycin did not correlate with the degree of vancomycin resistance in *E. faecium*. Daptomycin has bactericidal activity and a good safety profile, whereas linezolid is a bacteriostatic antibiotic associated with myelosuppression after two weeks of treatment. In Thailand, linezolid showed a high MIC against VRE species of *E. faecium* (MIC_50_ 1.5 μg/mL and MIC_90_ 2 μg/mL) [1,21], and it is difficult to achieve the target PK/PD of linezolid with standard dosing [25]. Hence, daptomycin is a potential treatment for VRE infection.

Before 2019, the daptomycin susceptibility breakpoint for enterococci at 4 μg/mL was higher than that of the *Staphylococcus* and *Streptococcus* breakpoint (1 μg/mL). This breakpoint was based on a standard dosing of 6 mg/kg every 24 h [26]. A Monte Carlo simulation study of the standard dose showed that MICs of 2–4 μg/mL could not achieve a PTA exceeding 90% of survival and the microbiological target of *E. faecium* [14,16], and daptomycin MICs of 3–4 μg/mL against *E. faecium* were associated with microbiological failure [27]. In 2019, the breakpoint of *E. faecium* was re-evaluated twice. The first revision changed the daptomycin-susceptible breakpoint to 1 μg/mL and the SDD was 2–4 μg/mL, with suggested dosings of 6 and 8–12 mg/kg, respectively. The second time, there was no susceptible category and the SDD category was a 8–12 mg/kg/day daptomycin dosing. The updated breakpoint revision achieved a target *f*AUC_0–24_/MIC of 12.9 for a 1-log_10_ CFU reduction in *E. faecium* in a murine model but not for the target in clinical outcomes [28].

The pharmacokinetics of daptomycin in critically ill patients changed from those of noncritically ill patients, with an increase in daptomycin clearance (0.4–0.6 to 0.9–1.05 L/h) and a slight increase in the volume of distribution (0.08–0.106 to 0.18 L/kg) [29,30,31]. Augmented daptomycin clearance leads to a lower daptomycin concentration and contributes to higher in-hospital mortality. In our study, the Monte Carlo simulation in critically ill patients at steady state produced results similar to previous data. A daptomycin dosing of 10–12 mg/kg/day can achieve 90% PTA of survival and microbiological outcomes at MIC ≤ 2 μg/mL, and a dosing of 8 mg/kg/day can achieve 90% PTA of microbiological target not for survival target at MIC ≤ 2 μg/mL [14]. At the first day, the %PTA of *f*AUC_0–24_/MIC was lower than the steady state. At MIC ≤ 2 μg/mL, administration of 12 mg/kg/day achieved 90% PTA of survival and microbiological target, whereas 10 mg/kg/day achieved only the microbiological target. No daptomycin dosing achieved 90% PTA of target at 4 μg/mL.

Several clinical studies have shown survival and microbiological eradication in patients with VRE bacteremia to be associated with the dose of daptomycin; by contrast, some studies have not found any benefit. The definition of a high dose varies [14]. Britt et al. showed that high daptomycin dosing (≥10 mg/kg/day) has lower 30 days mortality compared with medium and standard dosing (8 and 6 mg/kg/day, respectively), and both high and medium doses have good microbiological clearance compared with a low dose [12]. According to Chuang et al., higher daptomycin dosing (≥9 mg/kg/day) was correlated with lower 14 days mortality compared with low dosing (<7 mg/kg/day), regardless of the daptomycin MIC, but they did not find a difference in microbiological outcome [11]. In critically ill patients, we suggest a daptomycin dose of 8 mg/kg/day with a MIC of ≤1 μg/mL. Even though a 6 mg/kg/day dosing can achieve 90% PTA of MIC ≤ 1 μg/mL, concerns about treatment failure and higher mortality with a 6 mg/kg/day dosing should be considered. Mutations to the *liaFSR* system have been found [32]. At a MIC of about 2 μg/mL, we suggest a loading dose of 12 mg/kg followed by 10–12 mg/kg/day. At a MIC of 4 μg/mL, 12 mg/kg loading and maintenance doses with ampicillin or ceftaroline should be considered [33]. In our institution, the MIC value is low, and loading and maintenance doses of 8 mg/kg/day are optimal. 

There is an increased risk of skeletal muscle toxicity (myopathy and elevation of CPK) for daptomycin associated with daptomycin dosing and Cmin > 24.3 μg/L [10,17]. In the current study, the risk of musculoskeletal toxicity was low in this population, and a daptomycin dosing of up to 12 mg/kg/day was found to be safe. Multiple observational studies have shown that a ≥8 mg/kg/day dosing of daptomycin is not associated with CPK elevation and no serious adverse events were observed [11,12,14]. However, the risk of CPK elevation not associated with daptomycin dosing has been reported [11,30,34]. Thus, the CPK level should be closely monitored and use with statin is a concern.

There are several limitations of this study. First, several PK/PD targets of *E. faecium* were reported and the optimal target is unclear [14]. Daptomycin *f*AUC_0–24_/MIC > 27.4 is associated with improved survival in low-severity patients and not deep-seated infections [16]. Second, only 48 VRE were isolated for susceptibility testing. Further studies are needed to simulate daptomycin dosing for VRE infection in renal impairment patients.

## 4. Materials and Methods 

### 4.1. Microbiological Analysis

VRE strains were collected from each patient who was admitted to the Phramongkutklao Hospital, Bangkok, Thailand from October 2014 to February 2018 and met CDC/NHSN surveillance definitions for specific types of infections or sterile site specimens.

The procedures for in vitro activity were recommended by the CLSI. The vancomycin MIC was determined by broth microdilution (standard powder donated by Siam Pharmaceutical Co., Ltd.). The MIC of daptomycin was determined by E-test strips (Liofilchem, Teramo, Italy) with a range of 0.016–256 μg/mL. Incubations at 37 °C in ambient air were carried out for 18 and 24 h for daptomycin and vancomycin, respectively. Nonduplicate susceptibility testing of VRE was performed. The percentage of susceptible category was determined using the approved CLSI M100 30th edition breakpoint.

### 4.2. Pharmacokinetic and Pharmacodynamic Analysis and Dosing Simulations

Pharmacokinetic parameters were obtained from a previous study of critically ill patients [29]. Concentration versus time during the first 24 h and at steady state was determined using a one-compartment model. For pharmacokinetic and pharmacodynamic analysis, a 10,000-subject Monte Carlo simulation (Oracle Crystal Ball) was used to calculate the *f*AUC_0–24_/MIC > 27.4 and *f*AUC_0–24_/MIC > 20 for survival and microbiological response, respectively, and Cmin < 24.3 mg/L for safety. The simulation was conducted for various daptomycin dosing regimens (4, 6, 8, 10, and 12 mg/kg/day) and actual body weight values (55–65 kg).

The PTA was estimated at MICs of 0.25, 0.38, 0.5, 0.75, 1, 1.5, 2, 4, and 8 μg/mL and the CFR was calculated by %PTA against MIC distributions of VRE.

### 4.3. Ethical Approval

The Ethics Review Committee of the Royal Thai Army Medical Department, Bangkok, Thailand approved the study protocol (approval no. Q014b/62).

## 5. Conclusions

Based on the SDD breakpoint for enterococci and conditions with critically ill patients, daptomycin requires a dosing regimen of up to 8 mg/kg/day. Daptomycin resistance among VRE isolates was not observed in our study, and we found a low prevalence of VRE isolates with an SDD breakpoint. However, longitudinal or multicenter studies are needed to determine the real situation regarding VRE.

## Figures and Tables

**Figure 1 antibiotics-08-00245-f001:**
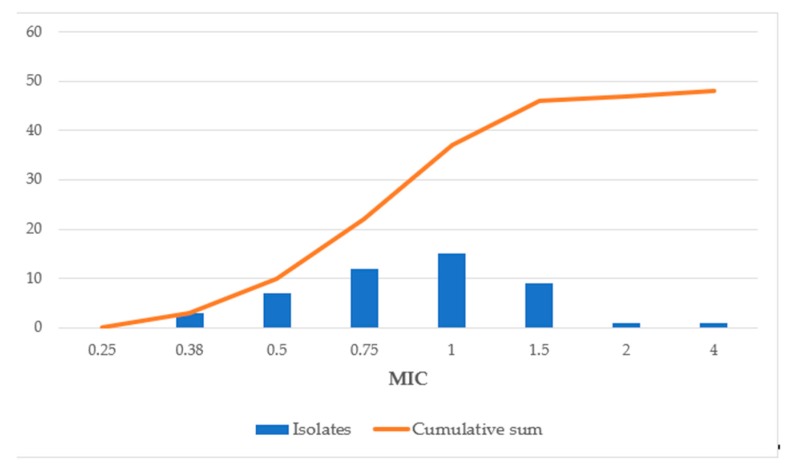
In vitro activities of daptomycin against 48 vancomycin-resistant enterococci (VRE) strains. **Note:** Nonduplicate susceptibility method.

**Figure 2 antibiotics-08-00245-f002:**
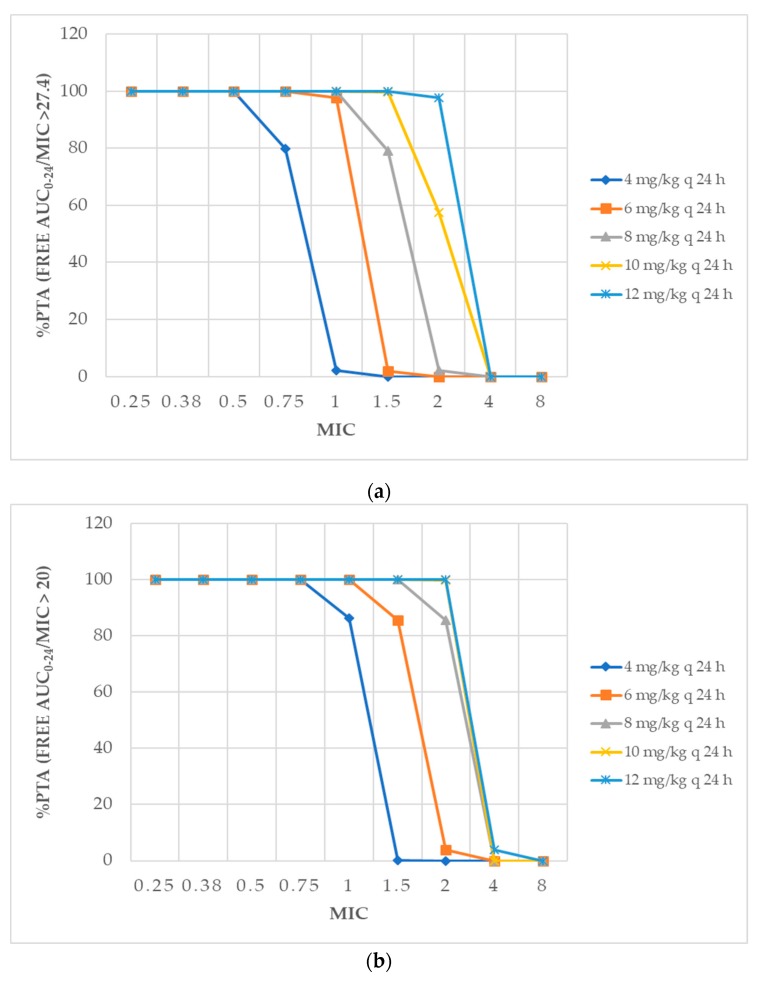
The percentage of probability of target attainment (PTA) for the different daptomycin dosings for critically ill patients during the first 24 h with targets of *f*AUC_0–24_/minimum inhibitory concentration (MIC) (**a**) > 27.4 and (**b**) > 20.

**Figure 3 antibiotics-08-00245-f003:**
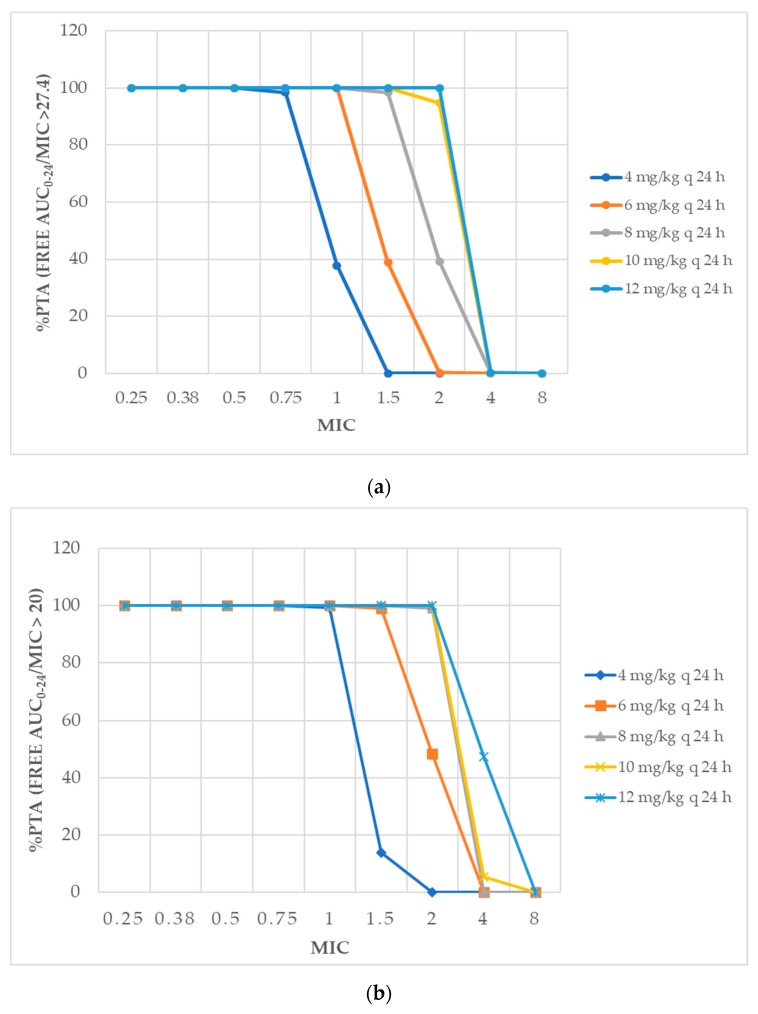
The percentage of PTA for the different daptomycin dosings for critically ill patients at steady state with targets of *f*AUC_0–24_/MIC (**a**) > 27.4 and (**b**) > 20.

**Table 1 antibiotics-08-00245-t001:** Cumulative fraction of response of daptomycin with various daptomycin regimens (%) in Phramongkutklao Hospital.

Daptomycin Dosing	% CFR of AUC/MIC				% Probability of Cmin < 24.3 mg/L
	>27.4		>20		
	First 24 h	Steady State	First 24 h	Steady State	
4 mg/kg q 24 h	41.47	57.24	72.86	79.42	100
6 mg/kg q 24 h	76.74	84.37	93.22	96.68	100
8 mg/kg q 24 h	91.98	96.35	97.61	97.9	100
10 mg/kg q 24 h	96.99	97.81	97.91	98.03	100
12 mg/kg q 24 h	97.87	97.92	98.0	99.0	99.99

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
