# Peer review of "Optimizing the Dosing Regimens of Daptomycin Based on the Susceptible Dose-Dependent Breakpoint against Vancomycin-Resistant Enterococci Infection"

_antibiotics, 2019, doi:10.3390/antibiotics8040245_

Round 1
Reviewer 1 Report
The authors followed appropriate methodologies (CLSI, CDC) to conduct this experiment. The findings are relevant to the hospital settings and informative to the clinical labs.
The English language need to be revised thoroughly. At several places it is difficult to follow the meaning of sentences (lines 55-57, 64-66, 84-86, 111-114, 120-125, among others).
Since the VRE strains were collected from inpatient, it will be valuable to know the resistance profile for the VRE isolates, MIC for vancomycin. The authors may discuss if there is an correlation among the vancomycin MIC and daptomycin MIC in these strains.
Please state if E-test strip results were confirmed by any other test. If not, was there any replication of the E-test performed on each isolate? Please mention it clearly in the methods and accordingly modify Figure 1.
Please replace 'critical ill patient' with 'critically ill patient' throughout the manuscript. Please also describe the strains with 2 and 4 mcg/ml MIC.
Reviewer 2 Report
This is a very interesting research article on the dosing of daptomycin to treat infections caused by vancomycin-resistant enterococci. The results are compelling and overall the article is well written. However, there are several parts of the text that are difficult to read. I would recommend that this article is subjected to a thorough proof-reading by a native English speaker. Finally, the presentation of the figures should improve. In particular, the style of Figures 2 and 3 is completely unrelated to Figure 1. It almost seems that Figure 1 was prepared for a different paper.
